# Changes in extreme regional sea level under global warming

S.-E. Brunnabend[1,2], H. A. Dijkstra[1], M. A. Kliphuis[1], H. E. Bal[3], F. Seinstra[4], B. van Werkhoven[4], J. Maassen[4], and M. van Meersbergen[4]

[1]Institute of Marine and Atmospheric Research Utrecht, Utrecht University, Princetonplein 5, 3584 CC Utrecht, The Netherlands.
[2]Leibniz Institute for Baltic Sea Research Warnemünde, Seestrasse 15, D-18119 Rostock, Germany.
[3]Department of Computer Science, VU University Amsterdam, 1081 HV Amsterdam, The Netherlands.
[4]Netherlands eScience Center, 1098 XG Amsterdam, The Netherlands.

*Correspondence to:* Sandra-Esther Brunnabend (sandra.brunnabend@io-warnemuende.de)

**Abstract.** An important contribution to future changes in regional sea level extremes is due to the changes in intrinsic ocean variability, in particular ocean eddies. Here, we study a scenario of future dynamic sea level (DSL) extremes using a high resolution version of the Parallel Ocean Program and generalised extreme value theory. This model is forced with atmospheric fluxes from a coupled climate model which has been integrated under the IPCC-SRES-A1B scenario over the period 2000-2100. Changes in 10-year return time DSL extremes are very inhomogeneous over the globe and are related to changes in ocean currents and corresponding regional shifts in ocean eddy pathways. In this scenario, several regions in the North Atlantic experience an increase in mean DSL of up to 0.4 m over the period 2000-2100. DSL extremes with a 10-year return time increase up to 0.2 m with largest values in the northern and eastern Atlantic.

## 1 Introduction

From satellite measurements, it has been well established that global mean sea level has increased by about 3 mm/yr over the period 1993-2010 (Rhein et al., 2013; Church and White, 2011; Church et al., 2013; Watson et al., 2015). However, regional sea level trends are very inhomogeneous over the oceans, and range from about a 10 mm/yr increase in the western tropical Pacific to about 5 mm/yr decrease in the subtropical eastern Pacific (Church et al., 2013). Regional deviations from global mean sea level rise occur due to ocean warming, global isostatic adjustment, land-ice mass loss and changes in the ocean circulation. The dynamic sea level (DSL) component is the sum of the contributions from local steric (thermal and saline) effects and ocean mass redistribution.

Until 2100, global mean sea level is projected to rise up to roughly one meter depending on the climate change scenario considered (Slangen et al., 2012, 2014). For example, under the SRES-A1B scenario, the global mean sea level is likely to rise between 0.42 m and 0.80 m (compared to 1986-2005), with major contributions provided by thermal expansion of ocean water and the mass loss of the major ice sheets and glaciers (Church et al., 2013; Slangen et al., 2014). For the highest radiative forcing scenario (RCP8.5), projected global sea level rise is between 0.52 and 0.98 m in 2100 (Church et al., 2013). Regional sea level changes projected for the North Atlantic show complex patterns that are partly caused by a weakening of the Atlantic Meridional Overturning Circulation (AMOC), by a shift in the path of the North Atlantic Current, and by changes in surface

buoyancy fluxes (Landerer et al., 2007; Yin et al., 2010, 2009; Pardaens et al., 2011; Kienert and Rahmstorf, 2012; Bouttes et al., 2013). Thermosteric sea level evolves with a pattern that reflects the reduced heat transport to the North Atlantic due to changes in ocean currents (Yin et al., 2010; Pardaens et al., 2011). For example, DSL is rising near the North American continent because of a reduction in the AMOC causing a redistribution of ocean mass (Landerer et al., 2007; Yin et al., 2009, 2010; Bouttes et al., 2013).

The spread in the projections of regional sea level change is largely determined by internal ocean variability and model uncertainty. In Slangen et al. (2012) and Bordbar et al. (2015), the spread due to decadal-to-centennial variability is considered by looking at ensemble simulations using CMIP3 and CMIP5 climate models, respectively. It was shown that the (CMIP5) ensemble spread of the projected DSL is of the same order of magnitude as the globally averaged sea level rise (Bordbar et al., 2015). Several regions were identified where the forced sea level change signal is relatively strong with respect to the internal variability, e.g. the Indo-Pacific part of the Southern Ocean and the eastern equatorial Pacific and hence may be detected earlier (Bordbar et al., 2015).

However, in all these model studies the strongest component of oceanic internal variability, i.e. that due to ocean meso-scale eddies, was not represented. Rectification processes due to eddies can lead to strong changes in mean ocean surface flows and their response to atmospheric forcing, in particular in the Southern Ocean (Böning et al., 2008). In strongly eddying ocean models even new modes of low-frequency variability may appear, such as the multidecadal Southern Ocean Mode (Bars et al., 2016). Using the eddy-permitting (about 1/4° horizontal resolution) version of the MIROC3.2 model, Suzuki et al. (2005) showed that representing ocean eddies provides a more detailed projection of regional sea level changes under the IPCC SRES-A1B scenario and that eddies are strongly involved in regional sea level extremes. In addition, as demonstrated by Firing and Merrifield (2004) from observational data, a high background sea level superposed on the sea level change due to an arriving ocean eddy can lead to extreme local sea levels.

Eddies can also have a strong effect on the sensitivity of the AMOC to freshwater forcing. The study of Weijer et al. (2012) indicates that the AMOC in the strongly eddying (about 0.1° horizontal resolution) version of the POP model version is more sensitive to freshwater perturbations than the non-eddying version of the same model. Climate model studies on the projections of the AMOC with non-eddying ocean components show only an AMOC decline of 22% to 40% over the period 2000-2100, depending on the IPCC scenario (Weaver et al., 2013). Only two (out of 30) of these models project a substantial decrease of the AMOC under the RCP8.5 scenario until year 2100 and no model shows an abrupt transition after the 21st century (Weaver et al., 2013). However, in high-resolution ocean models strong variations in the AMOC strength lead to changes in ocean currents and eddy pathways, which induce an additional contribution to the variability in DSL and hence affect extreme DSL values (Brunnabend et al., 2014).

It is important to assess the role of eddies in projections of future regional sea level changes, in particular on the DSL extremes. In this paper, we study a scenario of future DSL change using the high-resolution version of POP as in Brunnabend et al. (2014), but now forced with atmospheric fields from a coupled climate model that evolved under the SRES-A1B scenario. We focus on the changes in the probability density function of regional (and more local) DSL values and 10-year return extreme

values over the period 2000-2100, computed using the generalised extreme value theory (Coles, 2001), and compare these results to those obtained from a similarly forced non-eddying version of POP.

## 2   Ocean model

The high-resolution version of the POP used has a spatial resolution of $0.1°$ horizontally and 42 depth levels of which the thickness varies from 10 m near the surface to 250 m near the ocean bottom (Maltrud et al., 2008). The high spatial resolution captures the processes leading to meso-scale ocean eddies and provides a more detailed representation of the western boundary currents. Specific details about the high-resolution model setup, such as the treatment of the bottom topography, sea ice and river runoff, are described in the Weijer et al. (2012). The high-resolution model was optimised for use on the Cartesius supercomputer in Amsterdam (www.surfsara.nl) and about 3 model years are simulated per 24h using about 1000 cores.

The POP simulation was initialised from a 75 year spin-up simulation (Maltrud et al., 2008) under the CORE-I climatology dataset (Large and Yeager, 2004) as atmospheric forcing. This initial condition is indicated here as the year 1950. Under a freshwater flux which is diagnosed from the last five years of this spin-up, the model displays only a very small drift over a 200 year control simulation (Bars et al., 2016). Here, the model was forced with monthly mean atmospheric forcing fluxes over the period 1950-2100, that were derived from simulations with the ECHAM5-OM1 model within the ESSENCE (Sterl et al., 2008) project (see www.knmi.nl/~sterl/Essence/). The used forcing fields are 10 m wind speed, downward flux of short wave and longwave radiation, 2 m temperature, humidity, precipitation, runoff, and the surface wind stress field. The atmospheric forcing fields are given on a global $1° \times 1°$ grid and are interpolated to the curvilinear POP model grid. The outgoing heat and freshwater fluxes are computed within the model using bulk formulae. There is an initial adjustment after the switch in forcing in 1950, for example measured by the change in the AMOC strength, which lasts for about a decade.

Over the years 1950-2000, the POP model was forced by the ensemble mean atmospheric fields from the ESSENCE project that take observed concentrations of greenhouse gases and anthropogenic aerosols into account. Over the years 2001-2100, POP was forced with atmospheric forcing fields obtained from the ECHAM5-OM1 model according to the SRES A1B scenario (from an arbitrarily chosen ensemble member of the ESSENCE project (Sterl et al., 2008)). We focus on ensemble member #021 for which the high-resolution simulation is denoted by $R_{021}$. Two additional simulations are performed, using the forcing from the ESSENCE ensemble members #029 and #033 address the robustness the change of extreme DSL values.

In addition to this high-resolution simulation, a similarly forced simulation is performed with a low resolution POP version, indicated in the following by $R_{021}^{low}$. This non-eddying version has an average $1.0°$ horizontal resolution and 40 vertical levels (Weijer et al., 2012). The Gent and McWilliams (1990) scheme is used to represent eddy-driven tracer transports. Such a scheme is not needed in the strongly eddying version as these tracer transports are explicitly resolved.

The POP model directly computes the DSL, which can be decomposed into a mass redistribution term and a steric contribution. Because the freshwater flux is included into the model as a virtual salt flux and the global mean of precipitation, evaporation and river runoff is zero, no mass induced global mean sea level changes can be represented. Due to the applied

Boussinesq approximation, global mean steric sea level variations are not accounted explicitly during this study but this spatially independent contribution was computed from the model output (Greatbatch, 1994).

To demonstrate the performance of both versions of the POP model, we compare the DSL over the years 1993-2012 (computed from monthly means) with observations derived from altimetry. The altimeter products were produced by Ssalto/Duacs and distributed by Aviso, with support from Cnes (http://www.aviso.altimetry.fr/duacs/). No salinity or heat restoring is applied as even a weak salinity restoring is artificially constraining the AMOC. The model is also configured with no weak restoring of the global sea surface salinity field. However, as the POP model does not include a thermodynamic/dynamic sea-ice component, a prescribed climatological flux of heat and salt, is included in sea-ice region. These fluxes are the same for the control and hosing simulations and are an order of magnitude smaller than the mean fluxes. The sea-ice regions are indicated by white areas and are not considered in the analyses below. The mean DSL of simulation $R_{021}$ over the years 1993-2012 agrees well with observations, both for the mean (compare Fig. 1a and Fig. 1c) and the standard deviation (compare Fig. 1b and Fig. 1d). This shows that the model adequately determines the mean ocean circulation, including the western boundary currents and also represents the eddy-induced variability. Differences with respect to observations appear due to the general overestimation of the modeled variability during this time period, which may be due to the prescribed low resolution of the atmospheric forcing and the lack of feedback of the atmosphere on the ocean variability. Differences in variability may also occur due to the higher horizontal resolution of the ocean model ($0.1°$) than the altimetry dataset used ($0.25°$) as more small-scale features can be resolved. Regional differences in variability in the South Atlantic are caused by a too regular Agulhas ring formation rate in POP compared to observations (Bars et al., 2014).

In contrast, results for the low resolution simulation $R_{021}^{low}$ shown in Fig. 1e (mean) and Fig. 1f (standard deviation) indicate that only the mean DSL change is reasonable well captured. The variability captured in the model is mainly related to the seasonal cycle and internal variability is weak, in particular in the regions of the western boundary currents (similar to many other non-eddying ocean model results (Bordbar et al., 2015)). This weak variability also has consequences for the mean flow through the lack of representation of rectification processes causing, for example, too small DSL values in the Agulhas and the Gulf Stream regions.

## 3  Future dynamic sea level changes

In the results below, all long term changes are computed by taking the difference between values over the last 20 years (2081-2100) and the first 20 years (2001-2020) of the model simulations. Monthly mean data are used for the analysis of changes in the mean and standard deviation (section 3.1), while daily data are used in the extreme value analyses (section 3.2).

### 3.1  Mean and standard deviation

In the POP simulation $R_{021}$, global mean steric height increases by about 2.2mm/yr from year 2000 to 2100. As this signal is homogeneous over the Earth, it is not considered in the results below. Largest changes in mean DSL between the periods 2081-2100 and 2001-2020 occur in the North Atlantic (Fig. 2a), in particular near the western part of this basin (Fig. 2c). There

is a mean DSL decrease in the Atlantic and Pacific parts of the Southern Ocean, while mean DSL increases in the Indian part of the Southern Ocean. The mean DSL increases in the eastern part of the North Atlantic basin and decreases in the center of the subpolar gyre. Large changes in DSL variability occur in the Agulhas retroflection region and near Drake Passage (Fig. 2b). The DSL variability decreases in the western North Atlantic, in the center of the subpolar gyre and slightly along the western

boundary of the North Atlantic while it substantially increases in the eastern Atlantic (Fig. 2d). However, the separation of DSL change in the North Atlantic into steric height change and change in regional ocean mass show that the change is mainly caused by regional steric height changes (Fig. 3a,b). These regional steric height changes and the positive mass redistribution that increases DSL near the North American coast (Fig. 3c,d) correspond well with the pattern found by the study of Yin et al. (2009).

In the POP simulation $R_{021}^{low}$, global mean steric height varies only by a few cm over the period 2000 to 2100 and again is not considered further. Regional steric height changes and the redistribution of ocean mass towards the North American coast are also visible in the low resolution results. In addition, the small dipole pattern visible in the North Atlantic is caused by the reduced strength and the shift in ocean currents that are discussed later in this section. Regarding mean DSL patterns and amplitudes, the results of the low resolution simulation ($R_{021}^{low}$), as shown in Fig. 2e, agree well with many other model

studies using non-eddying ocean models (Landerer et al., 2007; Yin et al., 2009, 2010; Bordbar et al., 2015). At first sight, the results also look similar to those for the $R_{021}$ simulation (compare Fig. 2a and Fig. 2e). However, when regional details are considered, the results are different. The Southern Ocean basin contrast (Indian versus Atlantic/Pacific) is much stronger in the $R_{021}$ results. The DSL change in the Northern Atlantic is more dipolar in the North Atlantic than in the $R_{021}^{low}$ results, with a large area south of Greenland with decreasing mean DSL. The change in DSL variability is, as expected, different in both

models (compare Fig. 2b and Fig. 2f), in particular in western boundary current regions. In the North Atlantic, (compare Fig. 2d and Fig. 2h), the changes in variability are less coherent in the Gulf Stream region and have larger amplitudes in the eastern part of the basin.

To explain the changes in DSL in the North Atlantic for the $R_{021}$ simulation (Fig. 2c-d), the behavior of the Atlantic Meridional Overturning Circulation (AMOC) is shown in Fig. 4. The maximum AMOC at 26°N decreases from about 20 Sv

to about 5 Sv (red curve in Fig. 4a). The spatial pattern of the AMOC is not changing but the North Atlantic Deep Water is shallowing by about 1000 m (Fig. 4b-c). The maximum strength of the AMOC at 35°S decreases (blue curve in Fig. 4a) by more 60%. The decline in the AMOC causes a rise in mean DSL of up to 0.4 m near the North American continent, mostly because of a redistribution of ocean mass towards these regions (cf. Fig. 1a). The reduction of the AMOC in the $R_{021}^{low}$ simulation is only a few Sv, as in the ESSENCE ensemble (Sterl et al., 2008; Van Oldenborgh et al., 2009). The strong variations at 26°N

in the $R_{021}^{low}$ simulation is very likely due to an adjustment as a consequence of the change in forcing. At 26°N the AMOC measures also the Gulf Stream in the model, which can intensify due to a change in buoyancy gradient.

The DSL change in the Southern Ocean between the periods 2081-2100 and 2001-2020 (Fig. 2a) is caused by a southward shift of the westerly winds. In addition, the westerly wind stress strengthens by about 0.03 Pa (Fig. 6b). The increase in zonal momentum flux accelerates the Antarctic Circumpolar Current and increases the northward Ekman transport that changes the slope of the isopycnal surfaces in the South Atlantic (Yin et al., 2010). These effects cause changes in the water mass properties

leading to steric contraction in the Southern Ocean and steric expansion in the region of the Agulhas return current (Yin et al., 2010), explaining the results in Fig. 2a.

The reduction of the AMOC is also associated with a northward shift of the latitude separation of the Gulf Stream. This result has also been found in the non-eddying model studies (Landerer et al., 2007; Saba et al., 2016) and previous strongly eddying model studies (Brunnabend et al., 2014). In addition, eastward shifts of the path of the Gulf Stream and North Atlantic Current occur. This is shown more clearly by the change in surface mean kinetic energy (Fig. 5a) which has decreased over most of the Gulf Stream path in the $R_{021}$ simulation. Fig. 5c and 5d show the change of the eddy kinetic energy (EKE) of year 2090 with respect to 2010. The changes in the mean current path redirect eddies and lead to higher variability in the eastern Atlantic while in the sub-polar region the variability is reduced. In the $R_{021}^{low}$ simulation, similar shifts in the current system in the North Atlantic occur (Fig. 5b). However, the amplitude of the kinetic energy changes is much smaller compared to the $R_{021}$ simulation, in particular in the Labrador Sea and in the Caribbean Sea.

In the $R_{021}$ simulation, the global mean sea surface temperature (SST) rises by about 2°C over the period 2000-2100 (Fig. 6a). Almost all ocean regions experience a warming and near the east coast of North America, there is a warming of up to 4°C as also shown by Saba et al. (2016). However, in the Southern Ocean, SST remains almost unchanged over large regions. This can be explained by the atmospheric forcing fields associated with the SRES-A1B scenario as they lead to changes in the radiative forcing between atmosphere and ocean. In addition, SST is decreasing by more than 3°C in the subpolar gyre region of the North Atlantic. This cooling is related to changes in deepwater formation, as discussed by Weijer et al. (2012), associated with a decrease of the AMOC strength and the shift in the currents that reduce the heat transport to the northern polar regions, which leads to thermal contraction and a negative DSL change (Fig. 2a). The dipole pattern of SST changes and the corresponding changes in DSL are robust fingerprints of AMOC weakening and are consistent with most low-resolution coupled model projections (e.g. Drijfhout et al. (2012); Lorbacher et al. (2010); Danabasoglu et al. (2012); Drinkwater et al. (2014) and others).

The reduction of the AMOC also decreases the ocean-atmosphere temperature difference in the subpolar Atlantic region and hence leads to a reduction in the net ocean-atmosphere surface heat flux, i.e. a reduced heat loss to the atmosphere (Fig. 6c; positive values: flux into the ocean). However, this heat gain is not strong enough to compensate for the cooling caused by the reduced AMOC strength and the shift in current. The overall cooling in the subpolar gyre region in the North Atlantic tends to strength the AMOC but it cannot compensate for the influence of the general warming in the upper ocean. Furthermore, the cooling in this region leads to reduced evaporation resulting in a further freshening of the upper ocean (Fig. 6d) in a region where the AMOC is particularly sensitive to freshwater anomalies (Smith and Gregory, 2009; Weijer et al., 2012). The reduced heat loss and the additional freshening causes a further slowdown of the AMOC. The changes in surface fluxes for the simulation $R_{021}^{low}$ (not shown) are very similar as they are derived from the same atmospheric forcing fields, and are only slightly differently affected by the ocean fields, compared to the $R_{021}$ simulation. Because the mechanism of deep water formation is very different in the low-resolution model, the AMOC responds more mildly to changes in surface forcing than that in the high-resolution model (Weijer et al., 2012).

## 3.2 Regional PDF and Extremes

To determine an estimate of the Probability Density Function (PDF) of DSL we show histograms of modelled daily-mean DSL data over two 20 year periods (2001-2020 and 2081-2100). To remove variations on long time scales, all signals with frequencies lower than 550 days are first filtered out of these DSL time series. This leaves the seasonal and annual signals in the DSL time series and hence changes on these time scales also lead to changes in the PDFs and the DSL extremes. The PDFs are computed for three different regions in the North Atlantic, i.e. in the region of the subpolar gyre, near the US east coast and near the European coast (as shown in Fig. 7) using the daily-mean maximum value (over the region) in each of the regions from the daily-mean time series. The PDFs for three specific locations near the Azores, the Bermuda Islands, and Lisbon are also computed by using the monthly maximum value (at that location) from the daily time series. A Generalized Extreme Value (GEV) distribution function has been fitted to the PDFs using the maximum-likelihood method (Coles, 2001). It describes the behaviour of the extremes using the location, scale and shape parameterin (Coles, 2001) and is computed in the same way as in Brunnabend et al. (2014).

The changes in each PDF for the $R_{021}$ simulation for the different regions and locations are plotted in Fig. 8 with the blue histogram being the future PDF. The variance of DSL decreases in mid-Atlantic region 1 (cf. Fig. 2b,d), which is seen by the shift of the PDF to the left (Fig. 8a). This also leads to a reduction of the highest DSL extremes by more than 10 cm. In region 2 (western North Atlantic). DSL is mainly driven by mean changes due to steric effects and the mass redistribution and hence the PDF shifts to the right (Fig. 8c). In the eastern North Atlantic (region 3), the variance of the DSL increases (cf. Fig. 2b,d) due to the changes in the pathways of eddies causing the changes in EKE (Fig. 5c,d). This leads to a rightward shift of the PDF by about 10 cm in this region (Fig. 8e). The PDF of minimum DSL in region 2 and 3 (Fig. 8d,f) shifts left indicating an intensification of eddy activity affecting the sea level change in these regions. In region 1 (Fig. 8a), the PDF of minimum DSL shifts right as the intensity of the eddy activity decreases in this region.

Changes in the pathways of eddies are also important when considering local DSL extremes. The Azores are located in a region of slightly decreased variability (Fig. 1b,d) due to reduced eddy-kinetic energy in this region, shifting the PDF slightly to the left (Fig. 8g). Near the Bermuda Islands the shift in the ocean currents leads to lower probabilities of higher sea level extremes. (Fig. 8i). The most interesting result, however, is shown in Fig. 8k for the coast near Lisbon. Due to the shift in the Gulf Stream and North Atlantic Current one would expect increased probabilities for high DSL values in this region. However, because these currents are not only shifted but also reduced in strength almost no changes in DSL extremes can be identified (Fig. 8k). As the influence of ocean eddies decreases when reaching coastal region, no clear signal in the eddy intensity change can be identified at the three coastal locations (Fig. 8g-l).

The changes in the PDFs for the $R_{021}^{low}$ simulation show quite a different behavior than those in the $R_{021}$ simulation for most regions and locations. While the relative shift in the mean is comparable for both models in the regions 1 and 2 (Fig. 9a,b), the amplitude is much smaller for $R_{021}^{low}$. For region 3 (Fig. 9c), the PDF has bimodal characteristics and hardly changes under climate change, in contrast to the change in the $R_{021}$ simulation (Fig. 8c). The PDF change for the Azores is the opposite (Fig.

9d) in both models due to the fact that the eastward shift in the Gulf Stream has no influence on ocean eddy paths in the $R_{021}^{low}$ simulation (Fig. 5b). The PDFs of the other two locations (Fig. 9e,f) show the same behavior as in the $R_{021}$ simulation.

From the fit of parameters in Generalized Extreme Value (GEV) distributions, the extremes DSL values for a return time of 120 months (10 years) over the period 2001-2020 and their changes over the different 20 year periods (2081-2100 and 2001-2020) of the $R_{021}$ simulation can be determined (Fig. 10). Over the period 2000-2020 higher extreme sea levels occur in regions of high variability, i.e. in regions of the major current systems such as the Gulf Stream and the Agulhas Current (Fig. 10a,c). Therefore, the regional pattern of changes in extreme sea levels for a return time of 10 years (Fig. 10b,d) reflects the changes in sea level variability as shown in Fig. 2b,d. Sea level extremes can increase by 50 cm near Tasmania. Furthermore, in the northern and eastern North Atlantic, sea level extremes with a 10-year return time will increase by up to 20 cm. A comparison of the PDFs and the DSL extremes (for the 10 year return time) using a 550-day filter and a 180-day filter (not shown) indicates that the change in DSL extremes are dominated by the change in short term variability caused mainly by the shift in the ocean currents changing the eddy pathways (Fig. 5c-d).

To show that the mechanisms leading to extreme sea level change under the SRES-A1B scenario are robust, Fig. 10e-h show the change of extreme DSL values for a 10-year return time of two additional high-resolution simulations forced by the ensemble members 029 and 033. The similar pattern in the change of the extreme DSL values indicate similar changes in behaviour of the AMOC, ocean circulation and DSL as in the $R_{021}$ simulation.

Changes in extreme sea level values are shown in Fig. 11 for the $R_{021}^{low}$ simulation. The amplitude of these extremes is much smaller, in particular in western boundary current regions (Fig. 11a) and in the Gulf Stream region (Fig. 11c). The low-resolution ocean model simulation leads to different extreme sea level projections in the northern North Atlantic (in particular, in the Labrador Sea and Barents Sea) than for the $R_{021}$ simulation. The sign of the change in sea level extremes is also different in the Caribbean Sea. This shows the importance of including an explicit representation of eddy processes into an ocean model when looking at regional projections of DSL.

## 4  Summary and Discussion

In this paper, we considered future dynamic sea level (DSL) changes using a strongly eddying ocean model forced by atmospheric fields according to an SRES A1B scenario. The results show that changes in local and regional PDFs (between the periods 2001-2020 and 2081-2100) are mainly due to changes in DSL variability on short time scales and therefore related to changes in the ocean eddy field. This can be deduced from both the changes in the eddy kinetic energy of the ocean surface velocity field and from a comparison of DSL changes in a non-eddying version of the same model. In the high-resolution model simulation, the changes in eddy pathways are caused by a strong decrease of the AMOC with simultaneous eastward shifts in the path of the Gulf Stream and the North Atlantic Current.

Our main result is that the pattern of 10-year return time DSL extremes (as shown in Fig. 10) are determined by changes in the ocean eddy field (Suzuki et al., 2005; Brunnabend et al., 2014). In the POP model, eddies can come within 100 km of the coast and their maximum sea surface signal is often strongly correlated with that at the coast. In some regions of the globe

these extreme DSL values can be up to 0.5 m which of the same order of magnitude as the mean DSL change. This shows the importance of internal ocean variability for regional extreme sea levels, not only on the longer time scales (Bordbar et al., 2015), but also on the shorter time scales (Firing and Merrifield, 2004). These finding agree well with the study of Kanzow et al. (2009) where is has been shown that the influence of eddies on SSH variability is strongly reduced near ocean boundaries, but may still be several centimeters.

Low resolution ocean/climate models are not capable of accurately representing these changes in extreme sea levels. Some low resolution model studies do capture a shift in ocean currents in case of a declining AMOC (Landerer et al., 2007; Yin et al., 2010, 2009; Pardaens et al., 2011; Kienert and Rahmstorf, 2012). However, the model resolution is not resolving DSL variability caused by ocean eddies, as the parameterisation of eddies in these models only affects the heat and salt transport in the models. Although the use of an eddy-permitting ocean/climate model (with a $0.25°$ horizontal resolution) already indicated the importance of resolving ocean eddies to accurately estimate future sea level variability (Suzuki et al., 2005), the western boundary currents usually do not have a correct separation behaviour in these models.

There are several caveats in this model study which may modify the results quantitatively but which do not affect the main message of this paper that strongly eddying models are important for regional future sea level change projections. First, the AMOC in the $R_{021}$ POP model simulation appears to be quite sensitive to freshwater anomalies and hence the scenario here may be quite an extreme one. Second, the usage of an ocean-only model with mixed boundary conditions, restoring conditions below sea-ice regions and atmospheric forcing fields from a climate model is restricting the capabilities of the model in simulating the coupled ocean-atmosphere interactions occurring in reality. However, it is expected that shifts in the ocean eddy fields would also occur in coupled models with strongly eddying ocean model components. Third, the model does not simulate many other processes causing regional and coastal sea levels changes (e.g. GIA, and gravity). Many of these processes would only effect the mean DSL values and not its variability. Hence, as a first approximation, these sea level changes can be added to the mean DSL values (Slangen et al., 2012, 2014) determined here. Finally, although we show robustness using a small ensemble it would be better to use a larger ensemble of simulations (Bordbar et al., 2015) to determine the effect of ocean initial conditions and to have better statistics on the extreme DSL values. The latter is still hardly feasible with the current computational capabilities.

We conclude from the results that when developing plans for adapting to future changing sea level, not only mean regional changes should be considered, although they may be substantial. Also the changes in variability should be accounted for, as with higher variability the probability of sea level extremes may increase. This in particular holds for the North Atlantic region where many areas are vulnerable to sea level rise.

*Acknowledgements.* This study was supported by the Netherlands eScience Center (NLeSC) through the eSALSA (An eScience Approach to determine future Sea-level chAnges) project. The simulations have been performed on the Cartesius supercomputer at SURFsara (https://www.surfsara.nl) through the project SH-243-13. This work was also partially funded by the Dutch national research program COMMIT. The altimeter products were produced by Ssalto/Duacs and distributed by Aviso, with support from Cnes

(http://www.aviso.altimetry.fr/duacs/). To get access to the POP model results please write an email to the corresponding author. We thank the 2 anonymous reviewers for their constructive comments that improved the manuscript.

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

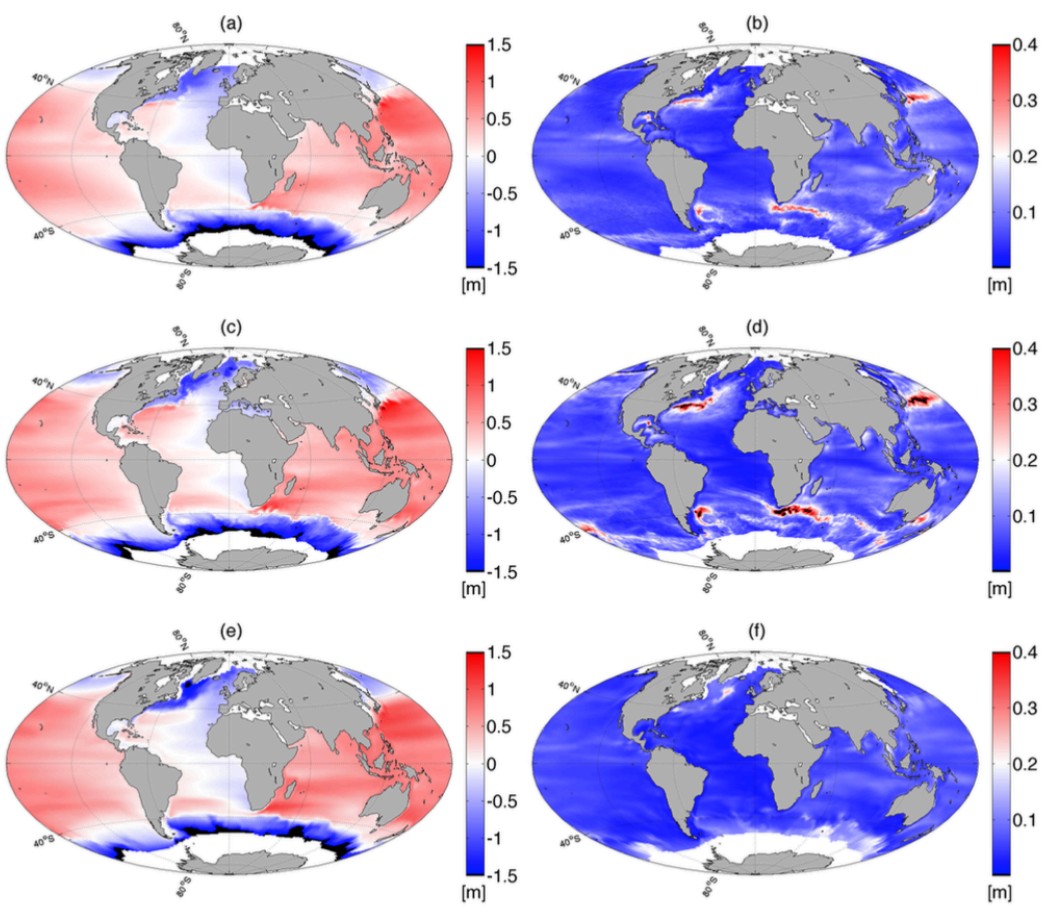

**Figure 1.** Mean sea surface height (SSH in meter) (a-c) and its standard deviation (b-d) over the years 1993-2012. (a,b) are derived from altimetry and (c-d) of the high-resolution simulation $R_{021}$. Panels (e,f) show the mean SSH and the standard deviation for the low-resolution simulation $R_{021}^{low}$, respectively.

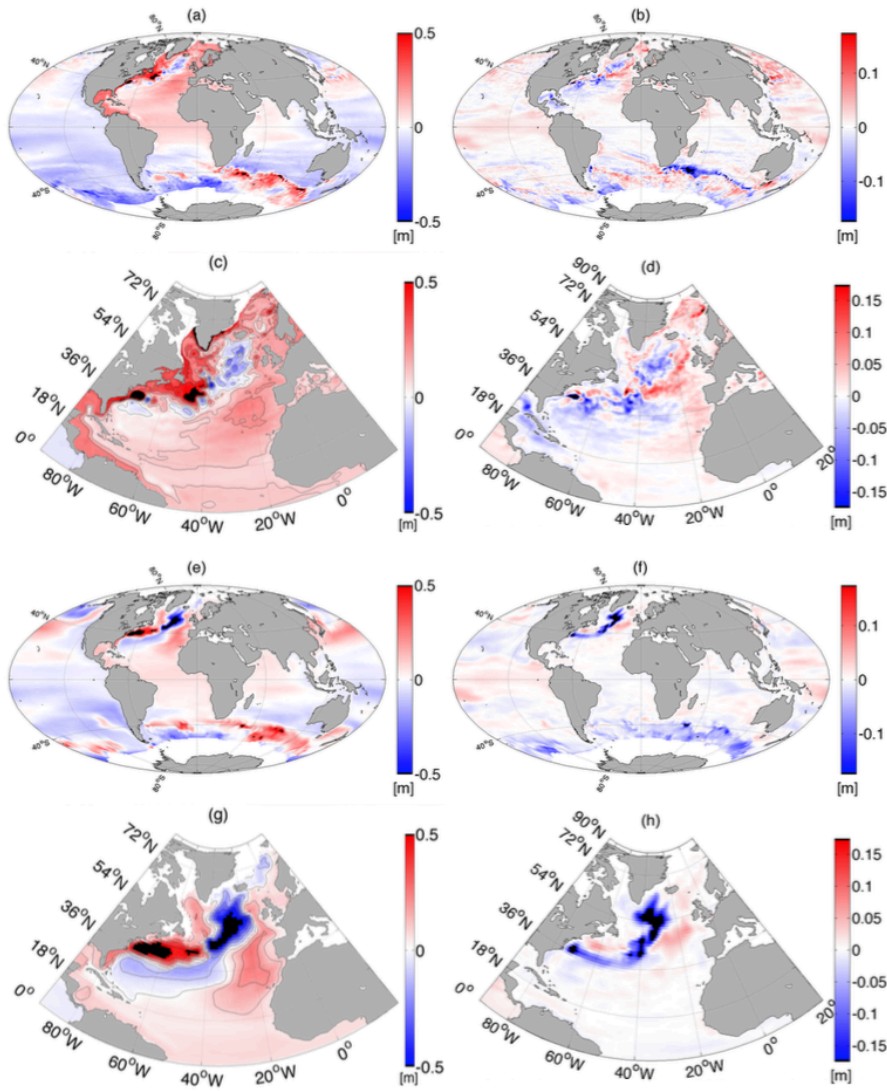

**Figure 2.** Change in the (a) mean and (b) standard deviation of modeled DSL in [m] between the periods 2081-2100 and 2001-2020 for the $R_{021}$ simulation. The panels (c) and (d) are magnifications of (a) and (b) for the North Atlantic region. (e-h) Same as (a-d), but for the $R_{021}^{low}$ simulation.

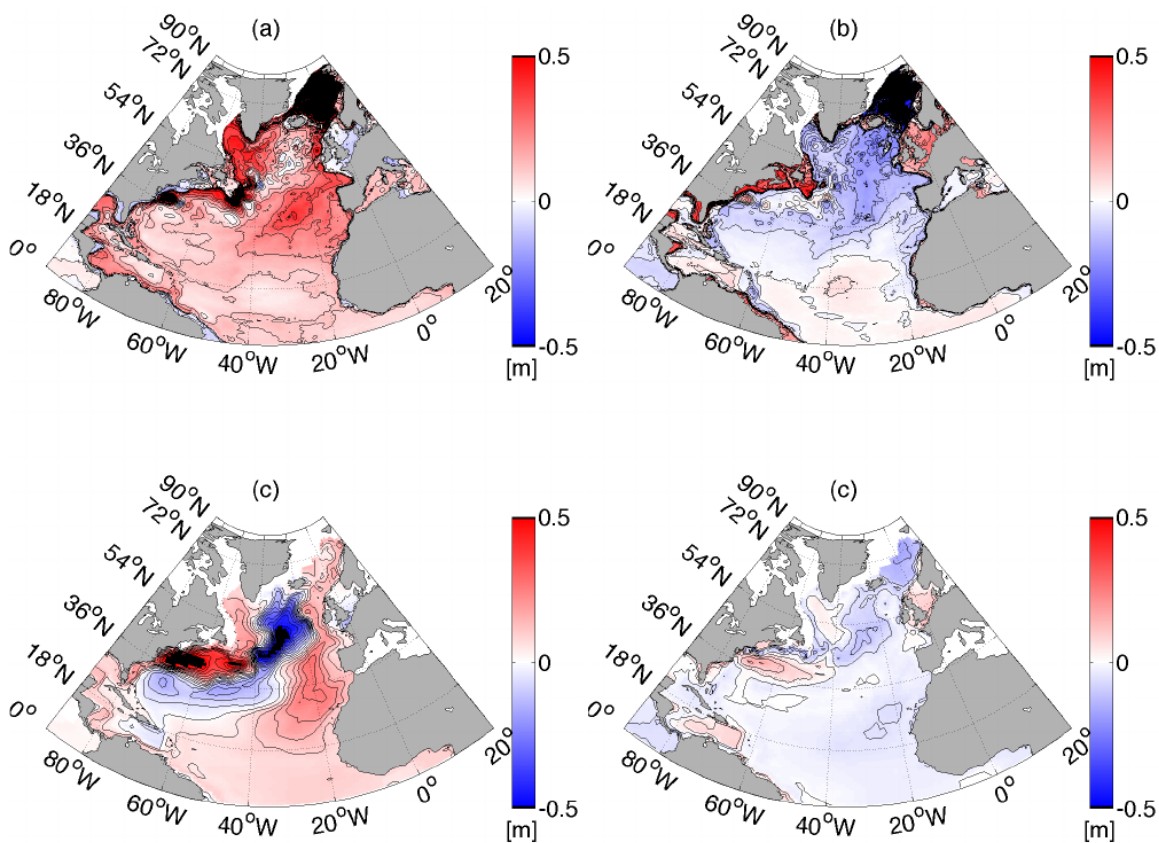

**Figure 3.** Change in (a) modeled mean steric height in [m], and change in (b) modeled mean ocean bottom pressure change in meter e.w.h. between the periods 2081-2100 and 2001-2020 for the $R_{021}$ simulation. The panels (c) and (d) are the same as (a,b), but for the $R_{021}^{low}$ simulation.

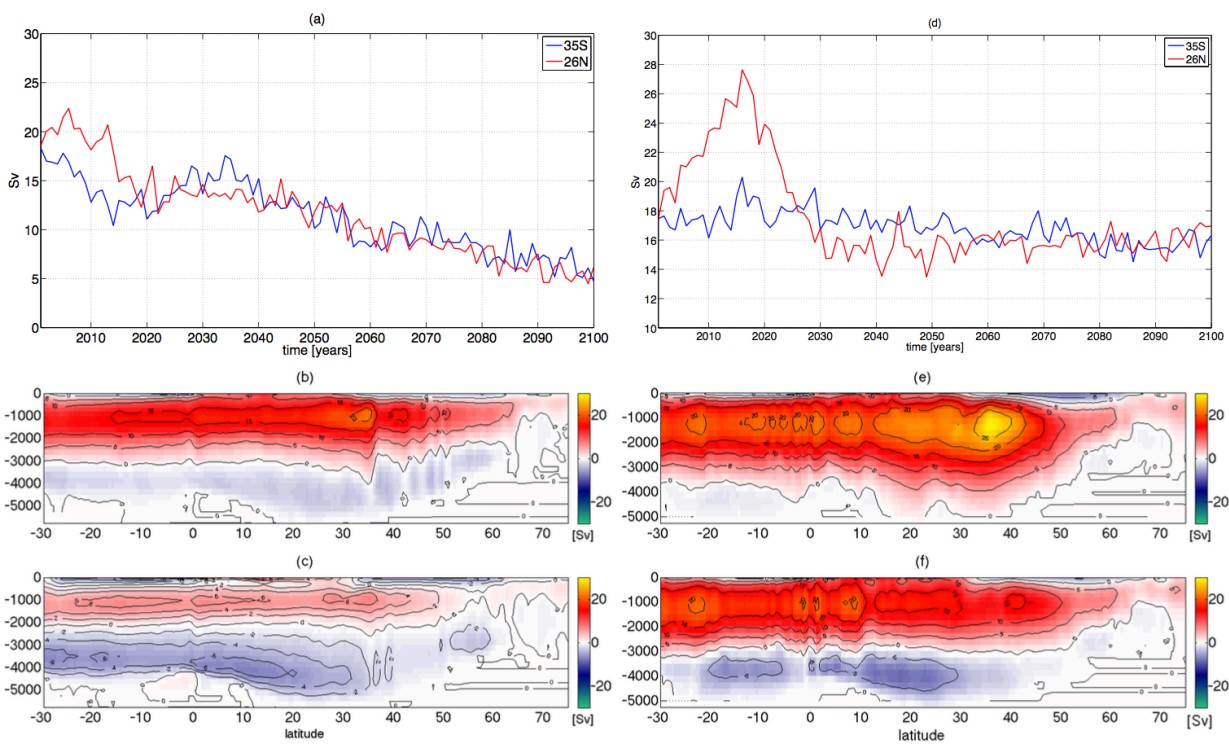

**Figure 4.** (a,d) Maximum AMOC strength at 35°S (blue) and 26°N (red) over the period 2000-2100 of (a-c) $R_{021}$ and (d-f) $R_{021}^{low}$; (b,e) Atlantic meridional overturning streamfunction (mean of years 2001-2020) (c,f) Same as (b,e) but over the period 2081-2100.

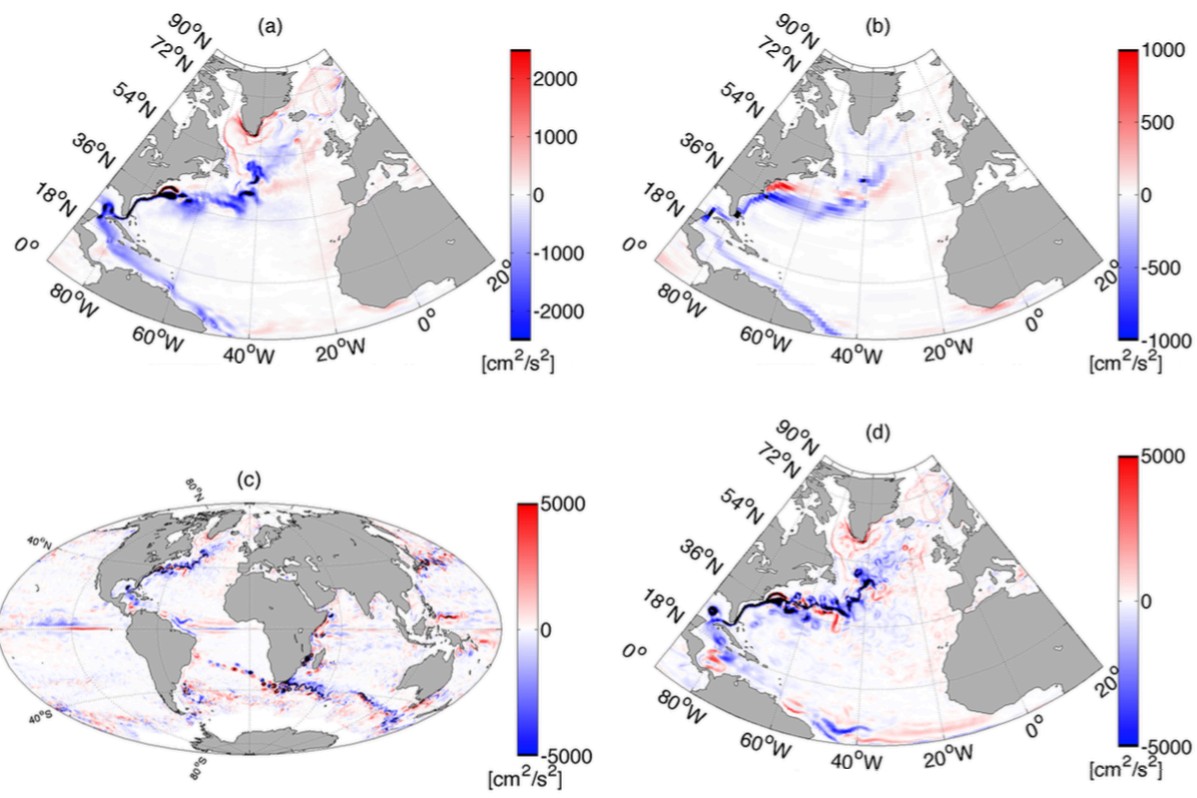

**Figure 5.** Difference of horizontal surface kinetic energy (energy flux per unit area) in $cm^2/s^2$ of the simulations (a) $R_{021}$ and (b) $R_{021}^{low}$ in the North Atlantic (mean of years 2081-2100 minus mean of years 2001-2020). (c) shows the difference in eddy kinetic energy (EKE) of the years 2090 and 2010 of $R_{021}$. Before computing EKE, the mean KE of the years 2080-2100 and 2000-2020 has been subtracted, respectively. (d) is the same as (c) showing only the North Atlantic.

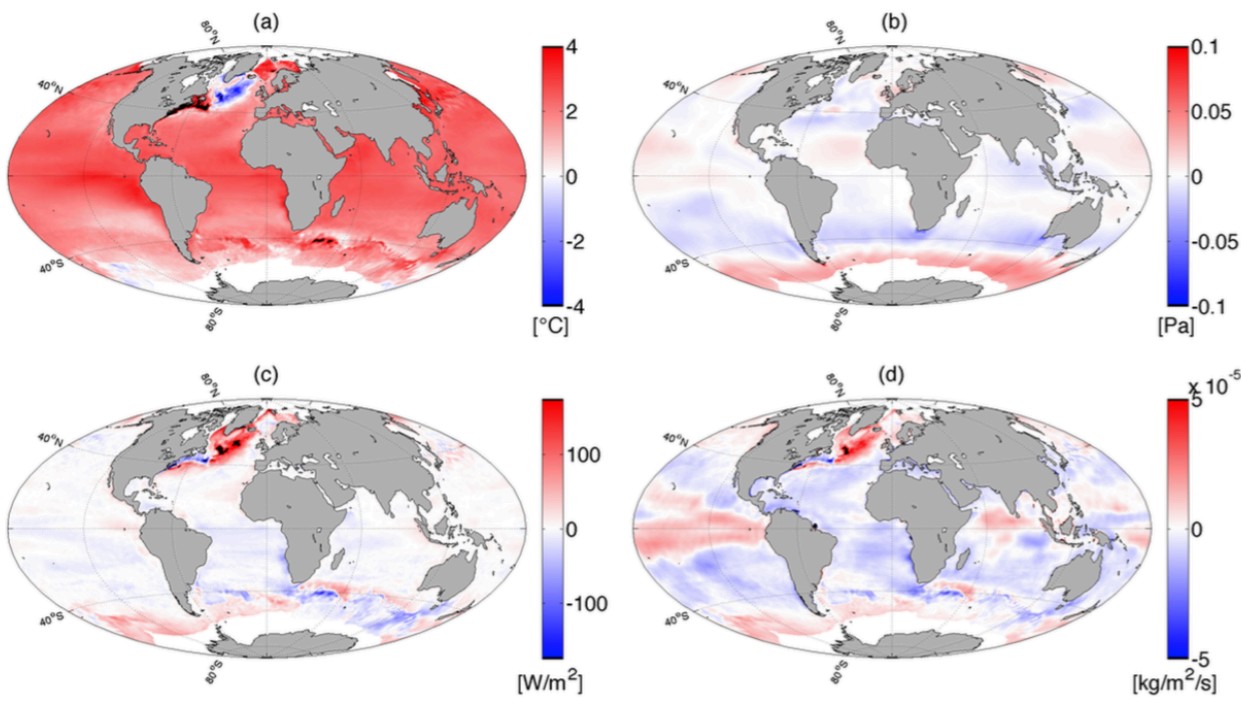

**Figure 6.** Change in (a) sea surface temperature ($^\circ C$), (b) zonal wind stress ($Pa$), (c) surface heat flux ($W/m^2$), and (d) surface freshwater flux ($kg/m^2/s$) for the $R_{021}$ simulation; again the mean over the last 20 years (2081-2100) minus that over the first 20 years (2001-2020) is shown. (positive values mean a flux into the ocean)

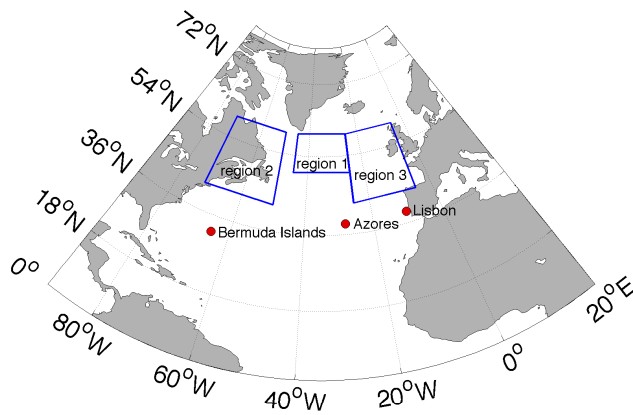

**Figure 7.** Regions in the North Atlantic ( region of the subpolar gyre, near the US east coast and near the European coast) and locations (near Losbon, Azores, and Bermuda Islands) used for determining the PDFs and for the extreme value analysis.

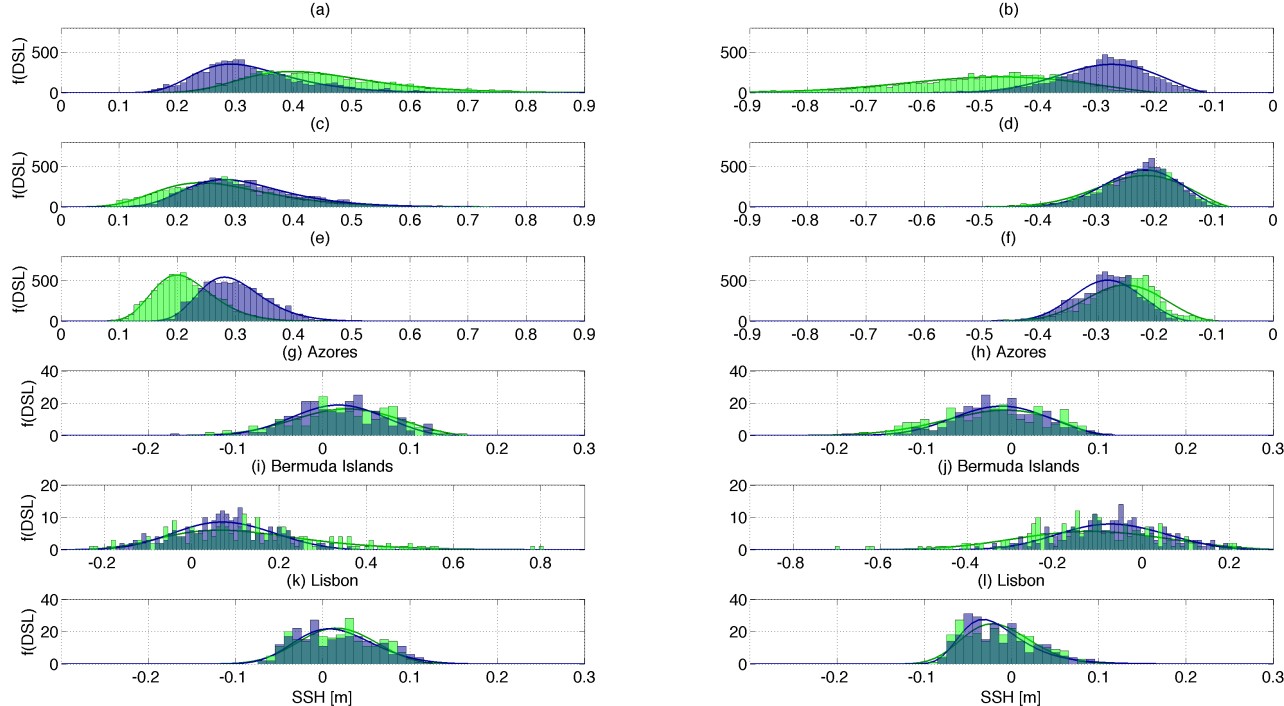

**Figure 8.** (a,c,e) Estimated Probability Density Function (PDF) of daily regional maximum DSL of simulation $R_{021}$ and (b,d,f) of the daily regional minimm DSL in the three different regions in the North Atlantic shown in Fig. 7 ((a): region of the subpolar gyre, (b): near the US east coast, and (c) near the European coast). In each plot, a maximum daily value over the region is identified after all variability with frequencies lower than 550 days has been filtered out. (g-l) Same, but for the locations indicated in Fig. 7 and using (g,i,k) monthly maximum local DSL values and (h,j,l) monthly minimum local DSL values derived from daily mean time series. The green histogram is the PDF for the first 20 years (2001-2020) and the blue histogram that for the last 20 years (2081-2100). The green and blue lines are the GEV distribution function fitted to the corresponding green and blue histogram, respectively.

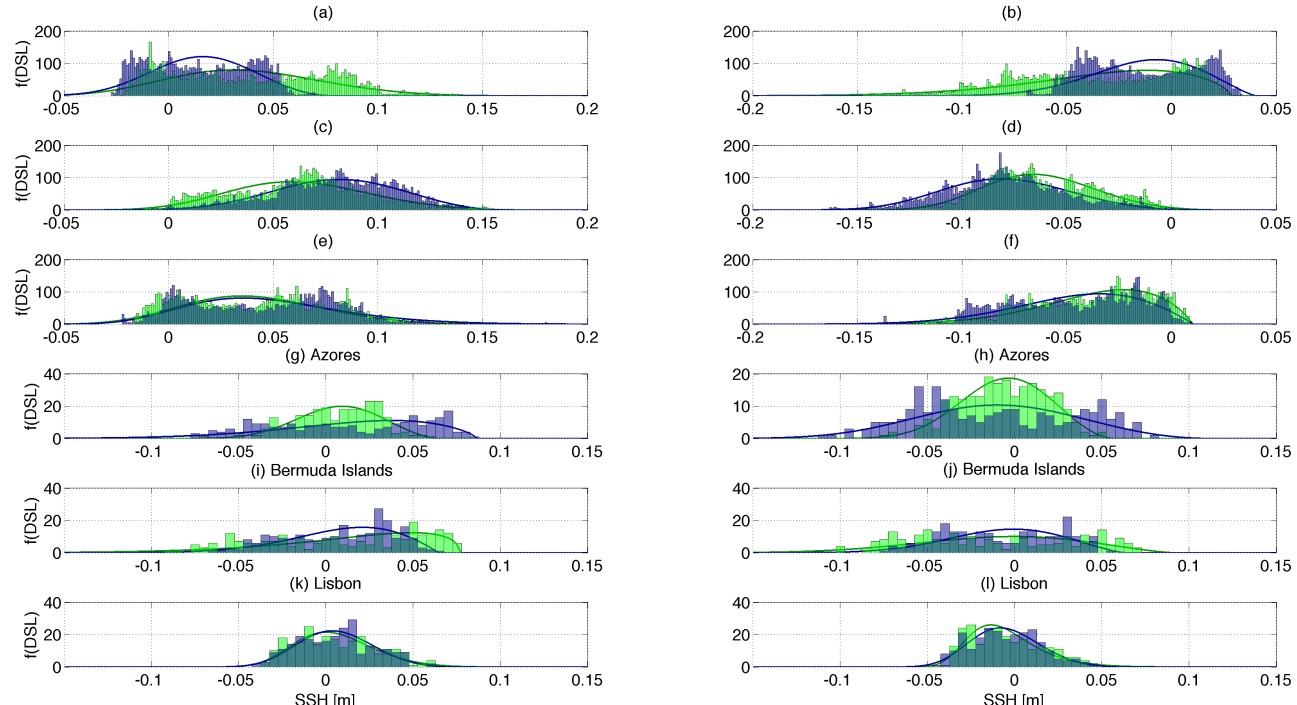

**Figure 9.** (a,c,e) Estimated Probability Density Function (PDF) of daily regional maximum DSL of simulation $R_{021}^{low}$ and (b,d,f) of the daily regional minimm DSL in the three different regions in the North Atlantic shown in Fig. 7 ((a): region of the subpolar gyre, (b): near the US east coast, and (c) near the European coast). In each plot, a maximum daily value over the region is identified after all variability with frequencies lower than 550 days has been filtered out. (g-l) Same, but for the locations indicated in Fig. 7 and using (g,i,k) monthly maximum local DSL values and (h,j,l) monthly minimum local DSL values derived from daily mean time series. The green histogram is the PDF for the first 20 years (2001-2020) and the blue histogram that for the last 20 years (2081-2100). The green and blue lines are the GEV distribution function fitted to the corresponding green and blue histogram, respectively.

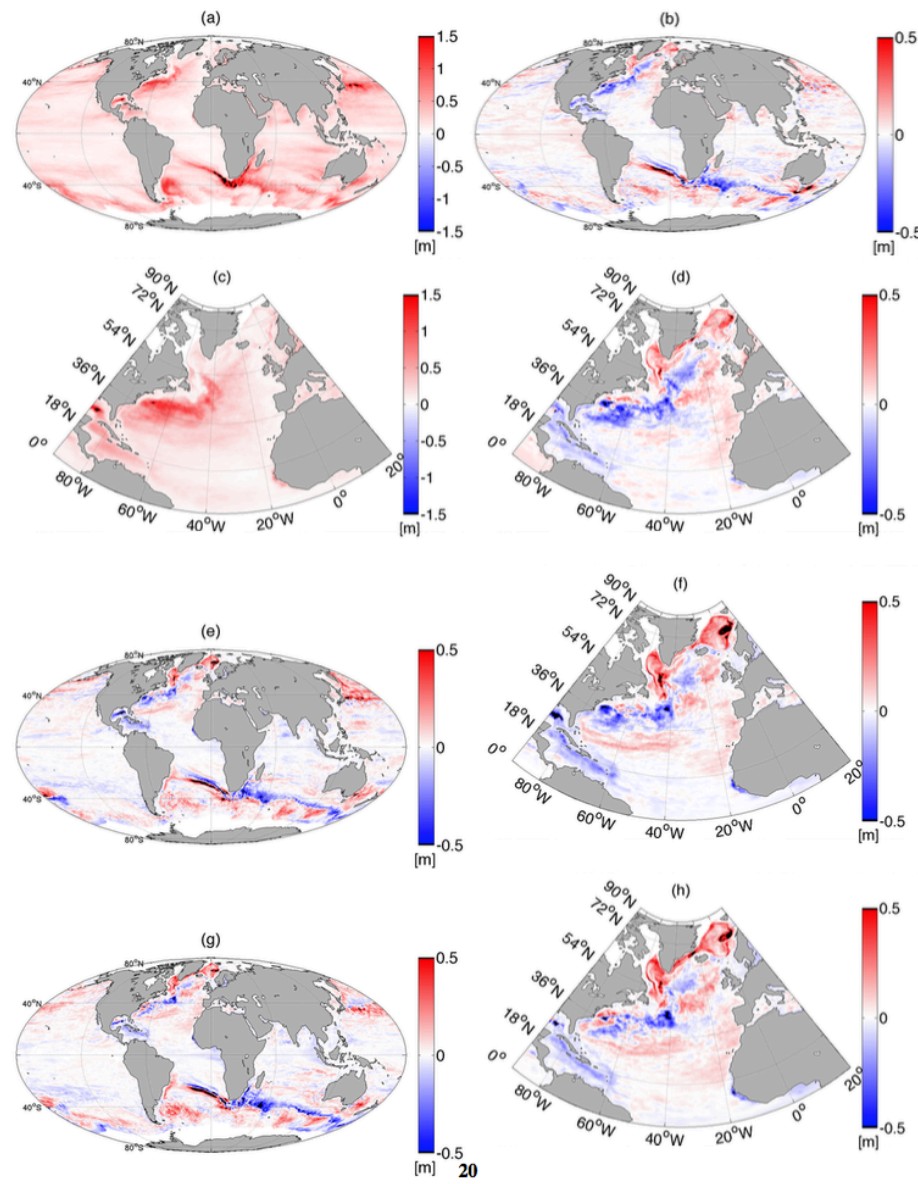

**Figure 10.** Extreme DSL values in [m] for a 10 year return time of simulation $R_{021}$ for (a) the first 20 years (2001-2020) and (b) the differences between the period 2081-2100 and 2001-2020. All signals with frequencies lower than 550 days have been filtered out. The panels (c) and (d) are magnifications of (a) and (b) for the North Atlantic region. (e,f) and (g,h) are the differences between the period 2081-2100 and 2001-2020 for two additional simulation forced by ensemble members 029 and 033, respectively.

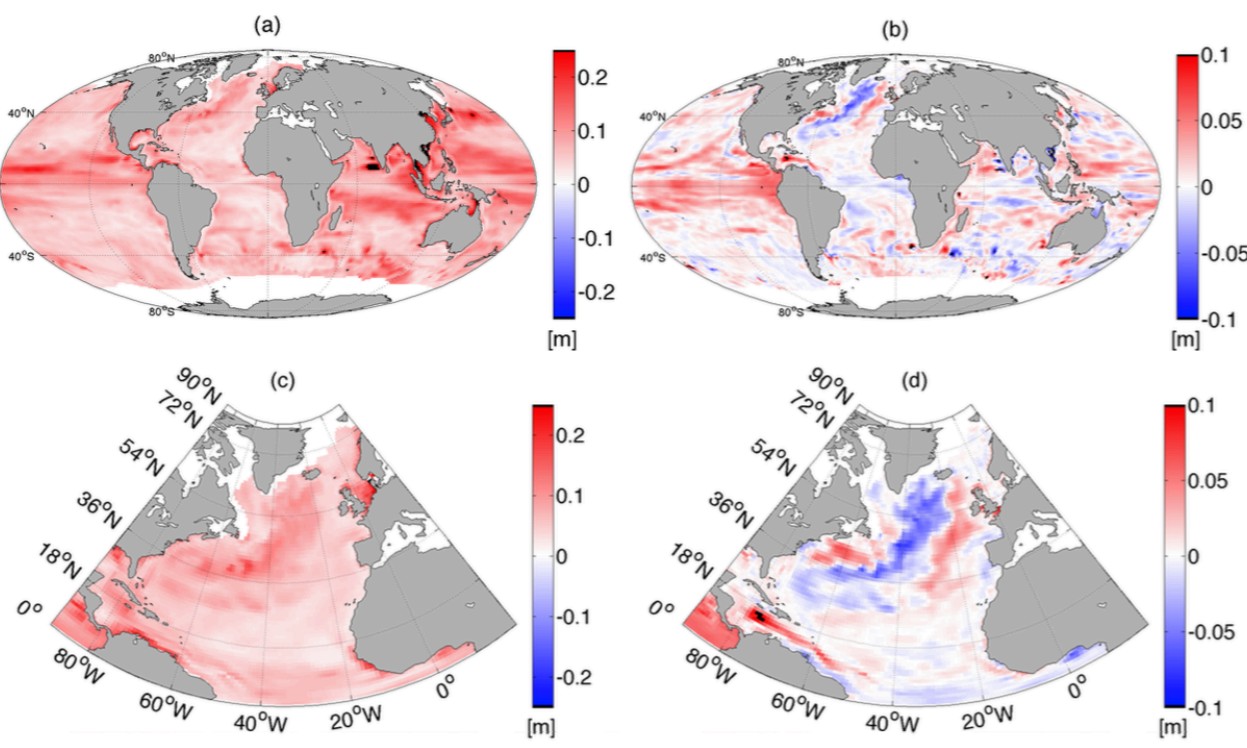

**Figure 11.** Extreme DSL values in [m] for a 10 year return time of simulation $R_{021}^{low}$ for (a) the first 20 years (2001-2020) and (b) the differences between the period 2081-2100 and 2001-2020. All signals with frequencies lower than 550 days have been filtered out. The panels (c) and (d) are magnifications of (a) and (b) for the North Atlantic region.