# Peer review of "Changes in extreme regional sea level under global warming"

_Ocean Science, 2016_

## Referee Comment (RC1) · Anonymous Referee #1 · 2 Sep 2016

The study focuses on projections of extreme dynamic sea levels associated with propagation of mesoscale eddies under future warming scenario. The model used here is an eddying ocean version forced with projected surface fluxes from another climate model. It is found that the change in dynamic sea level extremes is mainly caused by the change in ocean eddy pathway. In particular, both the mean and extremes of dynamic sea level in the North Atlantic show significant changes during the 21st century, with important implication of coastal impacts.

This is an interesting study, although the results are mainly based on ocean-only model simulations, in which the representation of air-sea feedback may not be complete. The mean and extremes of dynamic sea level anomalies are analyzed in combination, providing an overall picture about future sea level variability and change. I particularly like the comparison between the high and low resolution version of the model. It effectively

demonstrates the role of mesoscale eddies. The manuscript is written clearly.

So I recommend publication of the manuscript after a minor revision.

Line 17: global mean sea level "rise"

Line 76: Such "as"

Line 82: Does the model employ any salinity restoring at the ocean surface? It is known that the AMOC simulation is sensitive to different boundary conditions. The potential impact of the boundary condition on AMOC needs some discussion.

Line 111: Please provide reference for the altimetry data (1993-2012).

Line 159 and Figure 3: Figure caption should include model information. It would be better to add the AMOC time series from the low resolution model for comparison.

Line 164: Is it possible to show ocean bottom pressure changes?

Line 180: The paper by Saba et al. (JGR, 2016) should be cited for the northward shift of the Gulf Stream and the warming of the oceans near the US northeastern coast.

Line 181: This "warming hole" and dipole pattern of SST changes are robust finger-prints of AMOC weakening, consistent with most low-resolution coupled model projections.

Line 196 and Figure 5: Caption should indicate the direction of the heat and freshwater flux, that is, positive value means flux into the ocean.

Line 197: But cooling also increases density, which tends to strengthen the AMOC. Which process (more cooling vs less evaporation) is dominant?

Line 215 and Figure 7: The corresponding regions of the first three panels should be clarified further in the caption.

Figure 7: The shift of PDF is greater in the ocean interior but smaller along the coastal regions. I am curious about how often ocean eddies can actually approach coasts, or

they are mainly confined in the ocean interior. The potential impact of energetic eddies on coastal sea levels should be discussed further.

---

## Referee Comment (RC2) · Anonymous Referee #2 · 11 Oct 2016

The future sea level change is an important issue in an area that need investigating. The authors describe the future sea level change focusing on the extreme dynamical sea level change using eddy-resolving ocean model comparing with the lower resolution version. The results show that the changes in dynamical sea level extremes are mainly due to the changes in eddy pathways related ocean circulation changes. My only concern with the manuscripts is the lack of explanation of the statistical significant. This result is based on a single ocean circulation model under a single atmospheric condition, which is from results of a coarse resolution climate model. However, some global distinctive features are consistent with previous studies. The main points of the conclusions are plausible. They also use higher resolution model than previous studies. Therefore, I think these results are meaningful for further understanding of the extreme sea level change.

How about referring to use of the statistical method in abstract or introduction as high-resolution model. It makes it easy to understanding the following sections.

Line 60: Do you mean "slow shutdown beyond 2100"? Please clarify.

Line 97: The authors should refer the reason to choose the member. Are changes in ocean circulation, discussed in this manuscripts, seen adequately in the all ensemble members? If not, we should also consider the other mechanism of extreme sea level change.

Fig. 2: It would be better to add the contours of mean SSH on Fig. 2, that would make it easy to compare the sea level variability to the location of ocean current paths. Coarse resolution model cannot resolve mesoscale eddies. Is the variability seen in the low resolution simulation related to seasonal cycle?

Line 165: It would better to add the AMOC in low resolution simulation to Fig.3

Line 167: I feel that this sentence is not accurate. It is a kind of "Chicken or the Egg".

Line 180: Is this related to the NADW formation region in the model? Similar features also seen in the Low resolution version? Please add more information.

Line 219: The PDF shifts lefts in both region2 and region3. Does mean sea level rise affect the PDL shift in the region3 or changes in eddy pathway in the region3? Is it possible to show the PDF of minimum DSL? The PDF of minimum DSL could shift right if the intensification of eddy activity affects the sea level change in these regions.

Fig. 7: Blue and green lines indicate fitted GEV distribution? Please explain it.

―――――――――――――――――――

---

## Author Comment (AC1) · 10 Nov 2016

Below we provide responses to specific comments of Referee 1

Anonymous Referee #1 The study focuses on projections of extreme dynamic sea levels associated with propagation of mesoscale eddies under future warming scenario. The model used here is an eddying ocean version forced with projected surface fluxes from another climate model. It is found that the change in dynamic sea level extremes is mainly caused by the change in ocean eddy pathway. In particular, both the mean and extremes of dynamic sea level in the North Atlantic show significant changes during the 21st century, with important implication of coastal impacts. This is an interesting study, although the results are mainly based on ocean-only model simulations, in which the representation of air-sea feedback may not be complete. The mean and extremes of

dynamic sea level anomalies are analyzed in combination, providing an overall picture about future sea level variability and change. I particularly like the comparison between the high and low-resolution version of the model. It effectively demonstrates the role of mesoscale eddies. The manuscript is written clearly. So I recommend publication of the manuscript after a minor revision.

We thank the reviewer for the careful reading of the manuscript and the useful comments.

Line 17: global mean sea level "rise"

Corrected

Line 76: Such "as"

Corrected

Line 82: Does the model employ any salinity restoring at the ocean surface? It is known that the AMOC simulation is sensitive to different boundary conditions. The potential impact of the boundary condition on AMOC needs some discussion.

The POP model does not include a thermodynamic/dynamic sea-ice component. Therefore, a prescribed climatological flux of heat and salt is included in sea-ice region. However, no salinity restoring is applied outside these regions as even due to a weak restoring the AMOC is artificially constrained. This information is now added to the model description.

Line 111: Please provide reference for the altimetry data (1993-2012).

The altimetry dataset used is produced by Ssalto/Duacs and distributed by Aviso, with support from Cnes. The citation has been included in section 2 in addition to the citation in the acknowledgement.

Line 159 and Figure 3: Figure caption should include model information. It would be better to add the AMOC time series from the low-resolution model for comparison.

We included the model information in the caption and added the AMOC time series of the low-resolution model simulation, including a short discussion, in the manuscript.

Line 164: Is it possible to show ocean bottom pressure changes?

Yes, it is possible. A figure has been included into the manuscript (and discussed) showing mean and STD changes (similar to Fig. 1) in steric height and ocean bottom pressure, separately.

Line 180: The paper by Saba et al. (JGR, 2016) should be cited for the northward shift of the Gulf Stream and the warming of the oceans near the US northeastern coast.

We cited this interesting study when discussing the northward shift of the Gulf Stream and the warming of the ocean near the east coast of the US.

Line 181: This "warming hole" and dipole pattern of SST changes are robust finger-prints of AMOC weakening, consistent with most low-resolution coupled model projec-tions.

This issue is now discussed (with additional references) in the revised manuscript.

Line 196 and Figure 5: Caption should indicate the direction of the heat and freshwater flux, that is, positive value means flux into the ocean.

The direction of heat and freshwater flux has been added at both locations in the manuscript.

Line 197: But cooling also increases density, which tends to strengthen the AMOC. Which process (more cooling vs. less evaporation) is dominant?

The reduced heat loss leads to less cooling but cannot compensate for the overall cooling in this region caused by the reduced AMOC strength and the shift in ocean currents. The cooling also leads to less evaporation and a further freshening of the upper ocean. Therefore, both changes in atmospheric interaction lead to a further reduction of AMOC strength. However, the influence of the cooling in the subpolar

gyre region in the North Atlantic cannot compensate for the impacts of the general warming in the upper ocean. The corresponding paragraph in the manuscript has been reformulated to clarify this.

Line 215 and Figure 7: The corresponding regions of the first three panels should be clarified further in the caption.

The regions that correspond to the first three panels in figure 7 and 8 are now defined in the figure captions.

Figure 7: The shift of PDF is greater in the ocean interior but smaller along the coastal regions. I am curious about how often ocean eddies can actually approach coasts, or they are mainly confined in the ocean interior. The potential impact of energetic eddies on coastal sea levels should be discussed further.

Eddies can come within 100 km of the coast and their maximum sea surface signal is often strongly correlated with that at the coast

---

## Author Comment (AC2) · 10 Nov 2016

A point-by-point reply to all comments follows below.

Anonymous Referee #2 The future sea level change is an important issue in an area that need investigating. The authors describe the future sea level change focusing on the extreme dynamical sea level change using eddy-resolving ocean model comparing with the lower resolution version. The results show that the changes in dynamical sea level extremes are mainly due to the changes in eddy pathways related ocean circulation changes. My only concern with the manuscripts is the lack of explanation of the statistical significant. This result is based on a single ocean circulation model under a single atmospheric condition, which is from results of a coarse resolution climate model. However, some global distinctive features are consistent with previous studies.

The main points of the conclusions are plausible. They also use higher resolution model than previous studies. Therefore, I think these results are meaningful for further understanding of the extreme sea level change.

The authors thank Referee 2 for the useful comments on the manuscript. During the review process two additional simulations were finished and used to show that the change of extremes having a 10-year return time are robust.

How about referring to use of the statistical method in abstract or introduction as high-resolution model. It makes it easy to understanding the following sections.

The statistical method "generalized extreme value theory" is mentioned in the abstract and the introduction. In addition, "strongly eddying version" is changed to "high-resolution version" for clarification.

Line 60: Do you mean "slow shutdown beyond 2100"? Please clarify.

In the study of Weaver et al. 2012, two out of 30 of the investigated models project a substantial decrease of the AMOC under the RCP8.5 scenario until year 2100 and no model shows a abrupt transition after the 21st century. The sentence has been extended to make it clearer.

Line 97: The authors should refer the reason to choose the member. Are changes in ocean circulation, discussed in this manuscript, seen adequately in the all ensemble members? If not, we should also consider the other mechanism of extreme sea level change.

The member was chosen arbitrarily. However, during the time of review of the manuscript two more high-resolution simulations have been finished. To show that the mechanisms leading to extreme sea level change under the A1B scenario are robust, Fig. 9 is extended by showing the change of extreme DSL values for a 10 year return period for the other two simulations. From the similar results one can conclude that similar changes in behavior of the AMOC, ocean circulation and DSL occur. A

discussion is added to the manuscript.

Fig. 2: It would be better to add the contours of mean SSH on Fig. 2, that would make it easy to compare the sea level variability to the location of ocean current paths. Coarse resolution model cannot resolve mesoscale eddies. Is the variability seen in the low-resolution simulation related to seasonal cycle?

We have added these contour levels only in the panels c and g with a short discussion. Indeed, the variability in the low-resolution model is mainly related to the seasonal cycle as internal variability is weak. This is now mentioned in the revised manuscript.

Line 165: It would better to add the AMOC in low-resolution simulation to Fig.3

The figures corresponding to the low resolution results are now added to figure 3.

Line 167: I feel that this sentence is not accurate. It is a kind of "Chicken or the Egg".

The sentence is modified, where 'caused' has been changed to 'associated with'.

Line 180: Is this related to the NADW formation region in the model? Similar features are also seen in the low-resolution version? Please add more information.

Yes, the cooling is due to the changes in deepwater formation; we refer now to Weijer et al. (2012) who have discussed this.

Line 219: The PDF shifts lefts in both region2 and region3. Does mean sea level rise affect the PDL shift in the region3 or changes in eddy pathway in the region3? Is it possible to show the PDF of minimum DSL? The PDF of minimum DSL could shift right if the intensification of eddy activity affects the sea level change in these regions.

This is an interesting point, which is now addressed, in the revised paper.

Fig. 7: Blue and green lines indicate fitted GEV distribution? Please explain it.

The DSL extremes follow a Generalized Extreme Value (GEV) distribution. Therefore, one can fit a Generalized Extreme Value (GEV) distribution function whose parameters

(location, scale and shape) characterize the behavior of the extremes. We have added a short explanation to the caption of Fig. 7 and to the manuscript.

---

## Author Response (AR2)

Changes in extreme regional sea level under global warming

by S.-E. Brunnabend, H. A. Dijkstra, M. A. Kliphuis, H. E. Bal, F. Seinstra, B. van Werkhoven, J. Maassen, and M. van Meersbergen

submitted to the Ocean Science, doi: 10.5194/os-2016-57

**Reply to comments by the editor**

**Topic Editor Decision: Publish subject to technical corrections**
(17 Dec 2016) by Dr. Matthew Hecht, Comments to the Author:

The reviewers' comments have been addressed thoroughly, and I'm pleased to accept your paper, with one request: please verify that the model was configured with no weak restoring of the global sea surface salinity field. While the model is commonly run without such a weak restoring in fully coupled mode, it is unusual to do so when the atmospheric forcing is prescribed (whether reanalysis or derived from a coupled integration).

In any case, thanks for your good work in revision. Congratulations.

yours, --Matthew Hecht

Dear Matthew Hecht,

we thank you very much for accepting our paper. We verified that no weak restoring of the global sea surface salinity field has been applied to the simulations. Before the start of the simulations, the corresponding namelists had been configured to not include any weak or strong restoring of salinity or heat in the model. We added a sentence to the manuscript that no weak restoring of the global sea surface salinity field had been applied.

Yours sincerely,

Sandra-Esther Brunnabend

Changes in extreme regional sea level under global warming

by S.-E. Brunnabend, H. A. Dijkstra, M. A. Kliphuis, H. E. Bal, F. Seinstra, B. van Werkhoven, J. Maassen, and M. van Meersbergen

submitted to the Ocean Science, doi: 10.5194/os-2016-57

**Point-by-point reply to comments by the editor and referees**

*Dear Dr. Brunnabend, looking back over the two reviews, an important point will be to address the issue of statistical significance, and to clarify what one can robustly surmise from a strongly eddying model, with which one cannot feasibly produce ensembles or consider multiple forcings.*

*I look forward to seeing a revised version of the manuscript.*

*Yours, —Matthew Hecht*

Dear Matthew,

We thank you for pointing out the importance of the issue of statistical significance of changes in regional DSL extremes. We address this issue by extending Fig. 9 with results from two additional POP simulations, which use atmospheric forcings from two other ESSENCE ensemble members. From these results it can be concluded that the results in the original paper are robust.

Yours sincerely,

Sandra-Esther and Henk

Below we provide responses to specific comments of Referee 1 (original comments are grey/italic).

*Anonymous Referee #1*

*The study focuses on projections of extreme dynamic sea levels associated with propagation of mesoscale eddies under future warming scenario. The model used here is an eddying ocean version forced with projected surface fluxes from another climate model. It is found that the change in dynamic sea level extremes is mainly caused by the change in ocean eddy pathway. In particular, both the mean and extremes of dynamic sea level in the North Atlantic show significant changes during the 21st century, with important implication of coastal impacts.*

*This is an interesting study, although the results are mainly based on ocean-only model simulations, in which the representation of air-sea feedback may not be complete. The mean and extremes of dynamic sea level anomalies are analysed in combination, providing an overall picture about future sea level variability and change. I particularly like the comparison between the high and low-resolution version of the*

*model. It effectively demonstrates the role of mesoscale eddies. The manuscript is written clearly. So I recommend publication of the manuscript after a minor revision.*

We thank the reviewer for the careful reading of the manuscript and the useful comments.

*Line 17: global mean sea level "rise"*

Corrected

*Line 76: Such "as"*

Corrected

*Line 82: Does the model employ any salinity restoring at the ocean surface? It is known that the AMOC simulation is sensitive to different boundary conditions. The potential impact of the boundary condition on AMOC needs some discussion.*

The POP model does not include a thermodynamic/dynamic sea-ice component. Therefore, a prescribed climatological flux of heat and salt is included in sea-ice region. However, no salinity restoring is applied outside these regions as even due to a weak restoring the AMOC is artificially constrained. This information is now added to the model description.

*Line 111: Please provide reference for the altimetry data (1993-2012).*

The altimetry dataset used is produced by Ssalto/Duacs and distributed by Aviso, with support from Cnes. The citation has been included in section 2 in addition to the citation in the acknowledgement.

*Line 159 and Figure 3: Figure caption should include model information. It would be better to add the AMOC time series from the low-resolution model for comparison.*

We included the model information in the caption and added the AMOC time series of the low-resolution model simulation, including a short discussion, in the manuscript.

*Line 164: Is it possible to show ocean bottom pressure changes?*

Yes, it is possible. A figure has been included into the manuscript (and discussed) showing mean changes (similar to Fig. 1) in steric height and ocean bottom pressure, separately.

*Line 180: The paper by Saba et al. (JGR, 2016) should be cited for the northward shift of the Gulf Stream and the warming of the oceans near the US northeastern coast.*

We cited this interesting study when discussing the northward shift of the Gulf Stream and the warming of the ocean near the east coast of the US.

*Line 181: This "warming hole" and dipole pattern of SST changes are robust finger-*

*prints of AMOC weakening, consistent with most low-resolution coupled model projections.*

This issue is now discussed (with additional references) in the revised manuscript.

*Line 196 and Figure 5: Caption should indicate the direction of the heat and freshwater flux, that is, positive value means flux into the ocean.*

The direction of heat and freshwater flux has been added at both locations in the manuscript.

*Line 197: But cooling also increases density, which tends to strengthen the AMOC. Which process (more cooling vs. less evaporation) is dominant?*

The reduced heat loss leads to less cooling but cannot compensate for the overall cooling in this region caused by the reduced AMOC strength and the shift in ocean currents. The cooling also leads to less evaporation and a further freshening of the upper ocean. Therefore, both changes in atmospheric interaction lead to a further reduction of AMOC strength. However, the influence of the cooling in the subpolar gyre region in the North Atlantic cannot compensate for the impacts of the general warming in the upper ocean. The corresponding paragraph in the manuscript has been reformulated to clarify this.

*Line 215 and Figure 7: The corresponding regions of the first three panels should be clarified further in the caption.*

The regions that correspond to the first three panels in figure 7 and 8 are now defined in the figure captions.

*Figure 7: The shift of PDF is greater in the ocean interior but smaller along the coastal regions. I am curious about how often ocean eddies can actually approach coasts, or they are mainly confined in the ocean interior. The potential impact of energetic eddies on coastal sea levels should be discussed further.*

**Eddies can come within 100 km of the coast and their maximum sea surface signal is often strongly correlated with that at the coast (in the POP model).**

A point-by-point reply to all comments follows below. (original comments are grey/italic).

*Anonymous Referee #2*

*The future sea level change is an important issue in an area that need investigating. The authors describe the future sea level change focusing on the extreme dynamical sea level change using eddy-resolving ocean model comparing with the lower resolution version. The results show that the changes in dynamical sea level extremes are mainly due to the changes in eddy pathways related ocean circulation changes. My only concern with the manuscripts is the lack of explanation of the statistical significant. This result is based on a single ocean circulation model under a single atmospheric condition, which is from results of a coarse resolution climate model. However, some global distinctive features are consistent with previous studies. The main points of the conclusions are plausible. They also use higher resolution model than previous studies. Therefore, I think these results are meaningful for further understanding of the extreme sea level change.*

The authors thank Referee 2 for the useful comments on the manuscript. During the review process two additional simulations were finished and used to show that the change of extremes having a 10-year return time are robust.

*How about referring to use of the statistical method in abstract or introduction as high-resolution model. It makes it easy to understanding the following sections.*

The statistical method "generalised extreme value theory" is mentioned in the abstract and the introduction. In addition, "strongly eddying version" is changed to "high-resolution version" for clarification.

*Line 60: Do you mean "slow shutdown beyond 2100"? Please clarify.*

In the study of Weaver et al. 2012, two out of 30 of the investigated models project a substantial decrease of the AMOC under the RCP8.5 scenario until year 2100 and no model shows a abrupt transition after the 21st century. The sentence has been extended to make it clearer.

*Line 97: The authors should refer the reason to choose the member. Are changes in ocean circulation, discussed in this manuscript, seen adequately in the all ensemble members? If not, we should also consider the other mechanism of extreme sea level change.*

The member was chosen arbitrarily. However, during the time of review of the manuscript two more high-resolution simulations have been finished. To show that the mechanisms leading to extreme sea level change under the A1B scenario are robust, Fig. 9 is extended by showing the change of extreme DSL values for a 10 year return period for the other two simulations. From the similar results one can conclude that

similar changes in behaviour of the AMOC, ocean circulation and DSL occur. A discussion is added to the manuscript.

*Fig. 2: It would be better to add the contours of mean SSH on Fig. 2, that would make it easy to compare the sea level variability to the location of ocean current paths. Coarse resolution model cannot resolve mesoscale eddies. Is the variability seen in the low-resolution simulation related to seasonal cycle?*

We have added these contour levels only in the panels c and g with a short discussion.

Indeed, the variability in the low-resolution model is mainly related to the seasonal cycle as internal variability is weak. This is now mentioned in the revised manuscript.

*Line 165: It would better to add the AMOC in low-resolution simulation to Fig.3*

The figures corresponding to the low resolution results are now added to figure 3.

*Line 167: I feel that this sentence is not accurate. It is a kind of "Chicken or the Egg".*

The sentence is modified, where 'caused' has been changed to 'associated with'.

*Line 180: Is this related to the NADW formation region in the model? Similar features are also seen in the low-resolution version? Please add more information.*

Yes, the cooling is due to the changes in deepwater formation; we refer now to Weijer et al. (2012) who have discussed this.

*Line 219: The PDF shifts lefts in both region2 and region3. Does mean sea level rise affect the PDL shift in the region3 or changes in eddy pathway in the region3? Is it possible to show the PDF of minimum DSL? The PDF of minimum DSL could shift right if the intensification of eddy activity affects the sea level change in these regions.*

This is an interesting point, which is now addressed, in the revised paper.

*Fig. 7: Blue and green lines indicate fitted GEV distribution? Please explain it.*

The DSL extremes follow a Generalized Extreme Value (GEV) distribution. Therefore, one can fit a Generalized Extreme Value (GEV) distribution function whose parameters (location, scale and shape) characterize the behavior of the extremes. We have added a short explanation to the caption of Fig. 7 and to the manuscript.